# Wettability of Amino Acid-Functionalized PSMA Electrospun Fibers for the Modulated Release of Active Agents and Its Effect on Their Bioactivity

**DOI:** 10.3390/pharmaceutics15061659

**Published:** 2023-06-05

**Authors:** Sebastián Santander, Nicolás Padilla-Manzano, Bastián Díaz, Renato Bacchiega, Elizabeth Jara, Luis Felipe Álvarez, Cristóbal Pinto, Juan C. Forero, Paula Santana, Eugenio Hamm, Marcela Urzúa, Laura Tamayo

**Affiliations:** 1Departamento de Química, Facultad de Ciencias, Universidad de Chile, Las Palmeras 3425, Santiago 7800003, Chile; s.santander@ug.uchile.cl (S.S.); nicolas.padilla.m@ug.uchile.cl (N.P.-M.); bastian.diaz.j@ug.uchile.cl (B.D.); renato.bacchiega@gmail.com (R.B.); elizabeth.jara.c@ug.uchile.cl (E.J.); luis.alvarez.m@ug.uchile.cl (L.F.Á.); crpintog@ug.uchile.cl (C.P.); 2Escuela de Ciencias de la Salud, Universidad de Viña del Mar, Viña del Mar 2572007, Chile; juan.forero@uvm.cl; 3Instituto de Ciencias Aplicadas, Facultad de Ingeniería, Universidad Autónoma de Chile, El Llano Subercaseaux 2801, San Miguel, Santiago 8910060, Chile; paula.santana@uautonoma.cl; 4Departamento de Física, Facultad de Ciencia, Universidad de Santiago de Chile, Av. Víctor Jara 3493, Estación Central, Santiago 9160000, Chile; luis.hamm@usach.cl

**Keywords:** fibers, functionalization, amino acid, electrospinning, wettability, release

## Abstract

The ideal treatment for chronic wounds is based on the use of bioactive dressings capable of releasing active agents. However, the control of the rate at which these active agents are released is still a challenge. Bioactive polymeric fiber mats of poly(styrene-*co*-maleic anhydride) [PSMA] functionalized with amino acids of different hydropathic indices and L-glutamine, L-phenylalanine and L-tyrosine levels allowed obtaining derivatives of the copolymers named PSMA@Gln, PSMA@Phe and PSMA@Tyr, respectively, with the aim of modulating the wettability of the mats. The bioactive characteristics of mats were obtained by the incorporation of the active agents *Calendula officinalis* (Cal) and silver nanoparticles (AgNPs). A higher wettability for PSMA@Gln was observed, which is in accordance with the hydropathic index value of the amino acid. However, the release of AgNPs was higher for PSMA and more controlled for functionalized PSMA (PSMAf), while the release curves of Cal did not show behavior related to the wettability of the mats due to the apolar character of the active agent. Finally, the differences in the wettability of the mats also affected their bioactivity, which was evaluated in bacterial cultures of *Staphylococcus aureus* ATCC 25923 and methicillin-resistant *Staphylococcus aureus* ATCC 33592, an NIH/3T3 fibroblast cell line and red blood cells.

## 1. Introduction

A chronic wound can be defined as a type of wound that after a prolonged period of time has not been able to follow an optimal and timely regenerative process to generate an appropriate anatomical model and maintain the functional integrity of the affected site [1,2]. Within these types of wounds are pressure ulcers, diabetic foot ulcers and venous ulcers. Although Medicare has an estimated total cost of USD 28.1 billion per year for the care of these types of wounds [3], some studies claim that the total cost is still unknown [4]. Conventional wound treatment is based on the use of traditional commercial dressings, such as gauze, bandages (synthetic or natural) and compresses. These dressings need to be replaced regularly as they are easily moistened by wound drainage and tend to stick to the tissue, causing pain upon removal [5,6]. However, the ideal dressing is one that can adjust to the different characteristics of the wound, such as excess or absence of moisture or exudate, presence of living or necrotic tissue, susceptibility to infection, or prolonged healing times, among others. The ideal dressing should also be able to release active agents in the amount and rate necessary for each type of wound [7,8,9].

Several techniques have been developed to obtain advanced dressings. Dressings of polymeric origin (synthetic or natural) are commonly fabricated via the electrospinning technique [10,11,12]. This technique allows for obtaining polymeric fibers and, therefore, obtaining mats permeable to oxygen and water vapor which also have a morphology capable of resembling an extracellular matrix that stimulates cell regeneration [13,14,15]. The electrospinning technique can be implemented with a wide variety of polymeric solutions and allows the incorporation of active agents inside or on the surface of the fibers. The active agents can be subsequently released to the wound site, carrying out their bioactive effect. In this step, the ability to modulate the rate at which the active agents are released is crucial. For instance, a steady and sustained release over prolonged periods of time is necessary for a slow-healing wound requiring less frequent dressing changes, while a rapid and explosive release is suitable for wounds susceptible to infection requiring immediate antibacterial treatment [16,17,18].

There are several factors that affect the rate of release of active agents from polymeric fibers, such as fiber diameter, morphology and orientation, and mesh porosity, among others. In general, a high ratio between the surface area and the volume of the nanofibers will drastically increase the porosity of the mesh; thus, a larger surface area in contact with the medium will allow a higher release of active agents contained in the fibers [19,20,21]. On the other hand, the chemical composition of the fibers associated with the presence of polar or apolar functional groups will influence their wettability properties. In this sense, the ability to modulate the release of active agents can be controlled by the wettability properties of the polymer. This property is directly linked to one of the mechanisms of controlled release of active agents used when working with non-degradable polymers that have little or no solubility in water. Prior to the release of active agents, it is crucial that the water molecules can diffuse into the polymeric matrix, permeating between the fibers and also between the polymeric chains. This stage allows for the solubilization of the active agents and their subsequent diffusion to the external environment [22]. Therefore, in hydrophilic polymers, release will be favored, while in hydrophobic polymers, it will be attenuated. Furthermore, during solubilization and diffusion, intermolecular interactions may establish between the active agent and the polymeric matrix which will also influence the release capacity of the active agent. Hence, the knowledge of the structural characteristics of both the matrix and the active agent is relevant to predict or comprehensively understand the release phenomenon [23,24].

Several active agents have been used in the development of bioactive dressings. Among the most commonly used agents with antibacterial properties are natural extracts, such as cinnamon oil, thymol, turmeric and oregano essential oil, while synthetic active agents include metal nanoparticles and metal oxides [25]. Among metallic nanoparticles, silver nanoparticles (AgNPs) have been noted to be excellent antibacterial agents for both Gram-positive and Gram-negative bacteria. The efficacy of silver nanoparticles at low doses and low cytotoxicity [26] has led to the development of several commercially available dressings containing AgNPs [27]. The incorporation of nanoparticles into PVA fibers, either inside or on the fiber surface, has shown similar Ag^+^ release profiles, promoted mainly by the hydrophilic property of PVA [28,29]. However, Ag^+^ release on hydrophobic polymer fibers is more controlled [30].

On the other hand, the use of healing agents contained in bioactive dressings has been limited to vitamins, growth factors, stem cells and, in some cases, natural compounds [31]. Among the natural compounds, the use of *C. officinalis* extract has been associated with a healing capacity [32]. *C. officinalis* extract is constituted of various organic compounds, such as flavonoids, triterpenoids and polyphenols. Buzzi et al. reported significant reductions in the recovery time of venous leg ulcers [33]. Rad and co-workers incorporated *C. officinalis* extracts into electrospun dressings of poly(ε-caprolactone), zein and gum Arabic, observing improved fibroblast adhesion and proliferation [34]. At the molecular level, *C. officinalis* extract has been reported to activate transcription factors such as NF-κB (a transcriptional factor related to growth, cell survival, and immune and inflammatory response) and to increase the production of chemokine IL-8 in keratinocytes, which promotes migration and adhesion of monocytes and neutrophils in endothelial cells [35]. Several polymers have been used for making wound dressings via the electrospinning technique; among them, polylactic acid (PLA), polyvinyl alcohol (PVA), polyurethane (PU), chitosan (CS), collagen and alginate have been the most commonly used. [36,37,38]. In contrast, the use of copolymers such as poly(methyl vinyl ether-*alt*-maleic anhydride) [39], poly(methyl methacrylate-*b*-methacrylic acid) (PMMA-*b*-MA) [40], and poly(styrene-*co*-maleic anhydride) [41], have been less explored for the elaboration of wound dressings. Bearing in mind that the wettability property of polymeric fibers is important in the performance of a polymeric dressing, there are particular advantages for fibers elaborated from copolymers, for instance, their chemical versatility associated with greater availability of functional groups or their facility to anchor molecules of different polarities, which will allow for obtaining fibers with variable wettability. Poly(styrene-*co*-maleic anhydride) (PSMA) has been used to produce antibacterial polymeric fibers by functionalization with furanones [39], where the presence of the maleic anhydride group is of main interest, being able to participate in amidation reactions and conjugate with primary or secondary amines [42]. Polymers based on styrene maleic anhydride, such as PSMA, are used in multiple fields that require polymers capable of responding to stimuli, drug delivery, and biomedical applications, among others. This capacity is due to their unique structure, versatility, hydrophilic/hydrophobic character, and optical and catalytic activities, among others [43]. Previous research has demonstrated the bioactive properties of PSMA containing allantoin and AgNPs, showing its high ability to promote cell proliferation by increasing the growth rate of fibroblasts [44]. Corine and Hsieh [45] generated a polymeric matrix by electrospinning highly hydrophilic PSMA synthesized by a crosslinking method that allowed for the integration of hydrazine through the maleic anhydride group of PSMA. This process generated a great increase in hydrophilicity that was determined by an increase in the water absorption capacity and a significant reduction in the contact angle of the mats obtained by the researchers.

In this research, PSMA was modified with amino acids of different hydropathic indices. Tyrosine (Tyr), phenylalanine (Phe) and glutamine (Gln) were used to elaborate polymeric mats containing *C. officinalis* and AgNPs as active agents. In this work we hypothesize that the chemical functionalization of PSMA with amino acids of different hydropathic indices modulates the wettability properties of the mats and therefore the release rate of the active agents, thus influencing their bioactivity for bacterial cultures of *S. aureus* ATCC 25923 [SA], methicillin-resistant *S. aureus* ATCC 33592 [MRSA], NIH/3T3 fibroblast cell line and red blood cells. The results of contact angle and water absorption capacity measurements showed that the copolymer functionalization had an effect on the wettability properties of the mats, where the differences observed in the wettability, as well as in their structural characteristics, were mainly reflected in the Ag^+^ release behavior and antibacterial response.

## 2. Materials and Methods

### 2.1. Materials

Poly(styrene-*co*-maleic anhydride) (PSMA) Mw ~220,000 Da and *C. officinalis* 98% were purchased from BOC Science, while ACS Reagent silver nitrate ≥ 99.0%, oleic acid ≥ 99% (GC), sodium borohydride ReagentPlus^®^ 99%, sodium chloride ReagentPlus ≥ 99%, L-Glutamine ≥ 99% (TLC), L-Phenylalanine ≥ 98% (HPLC) L-Tyrosine ≥ 98% (HPLC) and Tween^®^ 80 were purchased from Sigma-Aldrich (St. Louis, MO, USA). *N*,*N*-dimethylformamide anhydrous 99.8%, and Dichloromethane (DCM) ACS Reagent ≥ 99.9% were obtained from Merck (Darmstadt, Germany). Deionized water with a resistivity of 18.2 MΩcm, conductivity of 0.055 μS/cm and a total organic carbon (TOC) content of less than 10 ppb was obtained from a LabStar 4-DI.

### 2.2. Functionalization of PSMA with Amino Acids

The functionalization of PSMA was carried out according to the scheme in Figure 1, using the amino acids L-Tyrosine (Tyr), L-Phenylalanine (Phe) and L-Glutamine (Gln), obtaining the functionalized polymers PSMA@Tyr, PSMA@Phe and PSMA@Gln, respectively. The reaction was carried out by solubilizing 3 g of PSMA and the corresponding amino acid in a 1:1 molar ratio in 80 mL of dimethyl sulfoxide (DMSO) at 80 °C and constant stirring in a nitrogen environment. Then, once the amino acid was completely solubilized, triethylamine (TEA) was added in an amount equivalent to 0.3% of the total moles. The progress of the reaction was monitored using an ATR-FTIR spectrophotometer (Shimadzu, IRSpirit, Kyoto, Japan). The spectra were performed at 24 °C with a relative humidity of 40%. The spectra were collected in the mats or reaction product by averaging 10 scans at 2 cm^−1^ resolution. 

### 2.3. Synthesis of AgNPs

AgNPs were synthesized by chemical reduction of silver nitrate (AgNO_3_) with sodium borohydride (NaBH_4_) in the presence of oleic acid as a stabilizing agent. Two solutions were prepared—a 5 mM AgNO_3_ solution and a 20 mM NaBH4 solution containing 0.528 mL of 75% oleic acid. The AgNO_3_ solution was added dropwise over the NaBH_4_ solution with oleic acid under constant stirring. The color change from translucent to dark brown confirmed the formation of AgNPs. The AgNPs solution was mixed with dichloromethane (DCM), then the immiscible mixture monopotassium phosphate [KH_2_PO_4_] was added as a transfer agent. Finally, the mixture was stirred until complete decolorization of the aqueous phase and brown coloration of the organic phase. The obtained organic solution was treated with a rotary evaporator to remove all the DCM [46].

### 2.4. Elaboration and Characterization of Polymer Mats

Polymeric solutions of PSMA and functionalized PSMA (PSMAf) were prepared at 30% *w*/*v* in a mixture of dichloromethane and dimethylformamide (DCM/DMF; 1:2 ratio, respectively). Then, Cal at 5, 10, 15, 15, and 20 wt.%, and AgNPs at 0.5, 1.0, 1.5, and 2.0 wt.% concentration were added to the former solutions. The electrospinning process was carried out using an electrospinning unit (TL-01) equipped with a needle with an internal diameter of 0.64 mm and a constant flow of 0.5 mL/h. During the electrospinning process, the collector was placed at a distance of 20 cm from the needle, with a temperature of 45 °C, relative humidity of 43% and a voltage of 27 kV. The mats obtained were characterized by field emission scanning electron microscopy (FESEM; FEI Quanta FEG250) operated at 15 kV to confirm the formation of fibers in the absence of deformations and to measure the diameter of the PSMA and PSMAf fibers. The samples were coated with thin films of 5 nm of gold prior to being observed.

The porosity and average pore diameter of the mats were quantified using Image J software version 1.54d (NIH, Bethesda, MD, USA) using 3 different images for each type of sample according to the threshold method [47,48].

### 2.5. Wettability of Polymer Mats

#### 2.5.1. Contact Angle

Contact angle measurements for each of the samples were performed on a contact angle device (Drop Shape Analyzer DSA25S, KRUSS, Hamburg, Germany) controlled by ADVANCE software version 1.5.1 (KRUSS). The contact angle on the surface of the mats was measured using the sessile drop method by depositing an 8 µL drop at 25 °C. Forward and backward angle measurements were performed to determine the contact angle hysteresis using Equation (1) [49].
∆θ = θa − θr(1)
where ∆θ corresponds to the contact angle hysteresis, θa to the forward angle, and θr to the reverse angle.

#### 2.5.2. Water Uptake

The PSMA and PSMAf mats were cut into 3 × 1 cm pieces and immersed in deionized water for a period of 1, 3, 5 and 7 days. After this time, the mats were removed from the water and the excess water (unbound moisture) was absorbed with a piece of filter paper. Water absorption was determined using Equation (2) [50].
% Water uptake = ((W_Wet_ − W_0_))/(W_0_) × 100(2)
where W_wet_ corresponds to the weight of the wet mat and W_0_ to the weight of the dry mat.

### 2.6. Release of AgNPs from Polymer Mats

The PSMA and PSMAf mats containing AgNPs as the active agent were cut into 4.5 × 4.5 cm pieces and immersed in a 2% solution of nitric acid (HNO_3_) for a period of 0.5, 1, 3, 5 and 7 days. The resulting solution was filtered twice to remove possible solid remains of the mat. For the first filtration, a 0.45 µm syringe filter was used and in the second filtration, a 0.22 µm syringe filter was used. Finally, the concentration of silver in the filtered solutions was measured by quadrupole mass spectrometry with an inductively coupled plasma source and a Thermo Scientific ICP-MS Q iCAP Q collision cell.

#### Analytical Methods for Ag^+^ Release Curves

The Ag^+^ ion release data from the different films were fitted to four kinetic models [51,52]: zero-order (Equation (3)), firsts order (Equation (4)), Higuchi (Equation (5)) and Korsmeyer–Peppas (Equation (6)).

Zero-order model:C(t) = C_0_ + K_0_t(3)
where C(t) is the Ag^+^ concentration in the medium at time t, C_0_ is the initial (zero) concentration, and K_0_ is the release kinetic constant, expressed in units of concentration per unit of time.

First-order model:C(t) = C_1_ (1 − e ^(−K_1_^^t)^)(4)
where C(t) is the Ag^+^ concentration in the medium at time t, C_1_ is the saturation concentration (unknown), and K_1_ is the release kinetic constant, expressed in units of time^−1^.

Higuchi model:C(t) = K_H_√t(5)
where K_H_ is the Higuchi dissolution constant. This model assumes Fickian diffusion kinetics.

Korsmeyer–Peppas model: C(t) = K_KP_ t^n^(6)
where K_KP_ is the Korsmeyer release rate constant, and n is the diffusion exponent or release exponent, which serves to characterize the different release mechanisms.

### 2.7. C. officinalis Release from Polymer Mats

The measurements of *C. officinalis* release over time were performed by immersing 1.0 × 2.0 cm pieces of the PSMA and PSMAf mats in 2 mL of deionized water. Samples were immersed for 0.5, 1, 2, 3, 4, 5 and 7 days, after which the absorbance of the solutions was measured in a UV-Vis spectrophotometer (Shimadzu UV-1900). To determine the concentration of Cal, a calibration curve was previously prepared, measuring the absorbance of the solutions at 281 nm. All measurements were performed in triplicate.

### 2.8. Antibacterial Test of Polymer Mats

Antibacterial assays were performed using the silver release solutions in deionized water on days 1 and 7 against *S. aureus* ATCC 25923 [SA] and methicillin-resistant *S. aureus* ATCC 33592 [MRSA]. Both bacteria were grown in a Tryptic Soy Broth (TSB) culture medium for 16 h at 35 °C. Subsequently, the bacteria were transferred to a fresh medium and allowed to incubate until the exponential phase. Subsequently, they were centrifuged at 6000 rpm for 2 min. Once the pellet was obtained, it was re-suspended in wash buffer (wash buffer: 9.7 mL sterile H_2_O, 100 µL sterile medium, 200 µL sterile PBS X1) and centrifuged again at 6000 rpm for 2 min (3 washes were performed). Finally, after the washes, both bacteria were standardized to 1 × 10^6^ CFU/mL by optical density at 600 nm.

Next, 20 µL of the standardized solution of bacteria at 10^6^ CFU/mL, 40 µL of the released AgNPs and 140 µL of wash buffer were deposited in a sterile Eppendorf tube for a total volume of 200 µL. The tubes were then left to incubate for 1 h at 35 °C with constant agitation. After this time, 10 µL of the incubated solution was taken and transferred to a new sterile Eppendorf tube containing 390 µL of wash buffer in order to dilute the sample. Next, 100 µL of the diluted sample was taken and seeded in a Mueller–Hinton culture medium. The cultures were incubated for 24 h at 35 °C. Finally, after 24 h, colony counting was performed. This assay was performed in triplicate on the mat for each day of release. In addition, culture plates were made in duplicate on each mat.

### 2.9. Cytotoxicity Assay of Polymer Meses

The cytotoxic response of mats was evaluated against an NIH/3T3 fibroblast cell line using the WST-1 Cell Proliferation Reagent (Sigma-Aldrich, Merck, Germany). For this assay, NIH/3T3 fibroblasts of murine origin were cultured in Dulbecco Modified Eagle’s Medium-F12 (DMEM-F12), with 10% fetal bovine serum, 100 U/mL penicillin and 0.1 mg/mL streptomycin (GIBCO, Grand Island, NY, USA) in tissue culture flasks. Once the fibroblasts were incubated, 5 × 10^4^ cells were transferred to a 96-well culture plate containing PSMA or PSMAf mats at the bottom of each well. Subsequently, it was left to incubate for 24 h in a cell incubator with 5% CO_2_ at 37 °C. After the incubation time, 10 µL of WST-1 reagent was added to 100 µL of cell culture and then poured onto each sample. Finally, the formation of colored formazan was assessed by reading the optical density at 470 nm using an ELISA plate reader. The following equation (Equation (7)) was used to determine cell viability [53].
Cell viability (%) = [OD_470 (experimental)_/OD_470 (non-active)_] × 100(7)
where experimental OD_470_ corresponds to the absorbance of the samples treated with PSMA or PSMAf mats with and without active agents. OD_470_ non-active is the absorbance of the sample treated only with PSMA and experimental OD_470_ corresponds to the absorbance of the samples treated with PSMA or PSMAf mats with and without active agents. OD_470_ non-active is the absorbance of the sample treated only with PSMA. For this experimental design, a sample treated only with PSMA was chosen as a negative control because the aim was to compare how PSMA functionalization with the presence of active agents affects cell viability. Triton X-100 was selected as a positive control for complete cell death.

### 2.10. Hemolysis Assay of Mats

For the hemolytic assay, 5 × 10 mm pieces of PSMA and PSMAf mats were cut with lime (concentrations 0, 5 and 20 wt.%) and AgNPs (0, 0.5 and 2.0 wt.%). In parallel, murine blood was extracted, washed and centrifuged 3 times at 2000 rpm for 20 min with PBSX1, obtaining a solution of erythrocytes with an approximate concentration of 6 × 10^8^ cells/mL. Then, each piece of mat was deposited in an Eppendorf tube and 100 µL of the 1% *v*/*v* erythrocyte solution and 30 µL of PBSX1 were added. On the other hand, a tube was prepared as a positive control triton X-100 1% *v*/*v* (100% hemolysis) and a second tube as a negative control PBSX1 (0% hemolysis). To both controls, 100 µL of the 1% *v*/*v* erythrocyte solution was added. Subsequently, all the samples were incubated for 1 h at 37 °C.

Next, each tube containing the sample and controls was centrifuged at 3000 rpm for 5 min. After this, 80 µL aliquots of the supernatant were taken and transferred to a 96-well culture plate and the absorbance was determined at 540 nm in a VERSA max microplate reader to detect free hemoglobin. Finally, the data were analyzed by determining the percentage of hemolysis according to the following equation (Equation (8)) [54].
Hemolysis (%) = [(A_540nm_ M) − (A_540nm_ PBSX1)/(A_540nm_ TritonX100) − (A_540nm_ PBSX1)] × 100(8)
where A_540nm_ of M corresponds to the absorbance value at 540 nm of the supernatant after the procedure, A_540nm_ PBSX1 corresponds to the absorbance of the negative control, and A_540nm_ corresponds to the absorbance of the positive control Triton X100.

## 3. Results and Discussion

### 3.1. Functionalization of PSMA with Amino Acids

The final product of the functionalization reactions of PSMA with the amino acids Gln, Phe and Tyr was characterized by ATR-FTIR spectroscopy in order to obtain information confirming the modification of PSMA with the different amino acids through the formation of an amide bond (Figure 2). In the PSMA spectrum, signals at 1779 cm^−1^ and 1854 cm^−1^, associated with the carbonyl groups of the maleic anhydride group are observed, while in the PSMAf spectra, these signals disappear. Instead, new signals at ~1703 cm^−1^ and ~1650 cm^−1^ appear. They correspond to the stretching of the carbonyl group of the carboxylic acid and the carbonyl stretching of the amide group, respectively. These signals account for the PSMA functionalization process according to the scheme in Figure 1. In addition, a signal is evident at ~1515 cm^−1^ for the PSMAf spectra, corresponding to the torsional vibrations of the N-H bond and the stretching of the C-N bond of the amide bond formed after functionalization. Finally, a signal is also observed in the PSMAf spectra at ~3449 cm^−1^, corresponding to N-H stretching. In the case of the PSMA@Tyr spectrum, a signal is evident at 1390 cm^−1^, which belongs to the bending of the hydroxyl in the Tyr residue. Additionally, the ATR-FTIR spectra of the PSMA and PSMAf mats in the absence and presence of the active agents of AgNPs (2.0 wt.%) and Cal (20 wt.%) were also obtained. The spectra are shown in Appendix A. For PSMA, the signal observed at 1643 cm^−1^ is attributed to the torsion of the C=C bond; however, a broadened signal is observed for the mats containing active agents, suggesting a possible interaction between the aromatic styrene ring present in the PSMA chain with the AgNPs and Cal. In the case of Cal, the interaction would be of the π–π type between the respective aromatic rings. In the case of PSMA@Phe in the presence of the active agents, a displacement of the signal by ~1000 cm^−1^ is attributed to the deformation of the C-H bond of a monosubstituted alkene present in the aromatic ring of Phe, indicating a possible interaction of the Phe ring with AgNPs and Cal. On the other hand, for PSMA@Tyr containing active agents, the presence of a broadened signal at 1700 cm^−1^ indicates a possible interaction of the carbonyl group of the ring opening with AgNPs and Cal. Finally, PSMA@Gln shows a signal displacement of 764 cm^−1^ suggesting a possible interaction of the amine group of the amino acid residue with AgNPs and Cal. 

### 3.2. Characterization of PSMA and PSMAf Mats

The polymeric solutions of PSMA, PSMA@Tyr, PSMA@Phe and PSMA@Gln were electrospun in order to obtain polymeric mats as shown in Figure 3. The SEM images reveal the presence of uniform and smooth fibers without bead formation, and the histograms show the fiber size distribution for each mat. According to the histograms, fibers made from PSMA@Tyr have the largest average diameter (3.0 ± 0.6) × 10^2^ nm, followed by PSMA@Gln (2.6 ± 0.5) × 10^2^ nm, and PSMA@Phe (2.0 ± 0.6) × 10^2^ nm. The fibers obtained from PSMA show a smaller diameter (1.6 ± 0.3) × 10^2^ nm compared to all PSMAf mats.

This increase in diameter could be due to steric hindrance generated by the large size of the amino acid residues or to repulsive interactions between the residues at the time of electrospinning, which would not allow a packing conformation as would occur in PSMA fibers without functionalization. The increasing diameter of the fiber due to steric hindrance between the polymer chains has also been described by Torricelli et al. [55].

On the other hand, since in the electrospinning process, the polymer is dissolved in a mixture of apolar/polar solvents (DCM/DMF), and at the same time it contains polar amino acids in its structure such as Gln and Tyr or apolar amino acids such as Phe, it is fundamental that the interaction of the latter with the solvent could determine the size of the fiber. This behavior has been reported in previous research by Erdem et al. [56], who performed tests with different combinations of a mixture of polar (tetrahydrofuran) and apolar (DMF) solvents. The researchers noticed that when electrospinning polyurethane, the fiber diameter increased as the polarity of the solvent increased (higher dielectric constant). The authors related the observed phenomenon to the transport of charges favored by a solvent with higher polarity. This same principle is applicable to the results obtained here, since by solubilizing a PSMAf with more polar groups, due to the incorporation of Gln and Tyr, the polymeric solution has a more polar character, thus favoring the movement of charges and generating larger fiber diameters.

Additionally, the mats prepared from the solutions containing active agents also showed variations in the size of the diameters. 

Figure 4 shows the SEM images of PSMA@Phe and histograms of the fiber diameter containing different concentrations of AgNPs, where an increase in diameter is observed as the concentration of AgNPs increases. This behavior could be due to the metallic and conductive character of AgNPs, which increase the conductivity of the solution and would generate an increase in the fiber diameter [47]. However, the PSMA@Phe 2.0 wt.% AgNPs do not follow this behavior, showing a reduction and higher variability in fiber diameter, probably due to the formation of large aggregates of AgNPs not homogeneously distributed along the fibers (see Appendix A), which would alter the conductivity of the polymer solution. 

The presence of AgNPs inside the fibers is shown in Figure 5. Image (A) shows the presence of nanoparticles (black dots) in PSMA@Phe fibers containing 0.5 wt.% AgNPs, with a homogeneous distribution of the nanoparticles in the absence of aggregates, while image (B) shows PSMA@Phe fibers containing 2.0 wt.% AgNPs, revealing the presence of clusters with sizes between 200 and 300 nm inside the fibers. A higher magnification image showing the morphology of the nanoparticles in more detail is observed in (C). Images (D) and (E) show FESEM images in the backscattering mode of PSMA@Phe fibers containing 1.0 and 1.5 wt.% nanoparticles, respectively; the white dots reveal the presence of AgNPs. Image (F) shows an X-ray energy dispersive spectrum, and the signal at ~3 keV associated with the binding energy of Ag_L_ confirms the presence of silver in image (E).

Figure 6 shows the SEM images of PSMA@Phe and histograms of the fiber diameter containing different concentrations of Cal. According to the histograms in Figure 6, an increase in fiber diameter is observed as the concentration of Cal increases. The increase in diameter could be due to the high content of phenyl groups with π orbitals in the Cal extract, which increases the conductivity of the solution [56]. On the other hand, this increase in fiber diameter could also be due to an increase in viscosity attributed to the presence of Cal, as indicated by Cramariuc et al. [57].

### 3.3. Wettability of Polymer Mats

#### 3.3.1. Contact Angle

Figure 7 shows the contact angle values for PSMA and PSMAf in the absence and presence of AgNPs. First, among the PSMAf mats without an active agent, a contact angle value of 136.8° is observed for PSMA@Phe, which is in agreement with the hydropathic index (HI) of the amino acid Phe (HI = 2.8), followed by the PSMA@Tyr and PSMA@Gln mats with a contact angle of 132. 7° and 28.7°, respectively, which is in agreement with the hydrophobic index value of Tyr (HI = −1.3) and Gln (HI = −3.5), where the PSMA@Phe mats are the most hydrophobic, while the PSMA@Gln mats the most hydrophilic. On the other hand, PSMA without functionalization presents an intermediate contact angle value of 130.4°. The contact angle values of the mats containing AgNPs show a slight decrease in the angle value, confirming the more polar character provided by the AgNPs compared to the mat in the absence of the nanoparticles.

On the other hand, in the mats containing *C. officinalis* (Figure 8), a tendency towards an increase in the contact angle is evident as the concentration of the active agent increases. This behavior is mainly associated with the apolar character of this type of extract. Appendix A show the values of advancing and receding angles for all the mats, where the difference between them provides the hysteresis value related to the chemical heterogeneity of the surface of the mats. From the tables, it is possible to notice that the hysteresis of the PSMA@Phe mats is lower compared to the rest of the mats, which indicates that the distribution of the active agent in these mats could be more homogeneous, followed by the PSMA mats, and then the PSMA@Tyr mats, with the latter being the most heterogeneous.

#### 3.3.2. Water Uptake Capacity of the Mats

Figure 9 shows the water absorption capacity plots for both PSMA and PSMAf. There is a significant variability in the absorption capacity of PSMA, while the absorption behavior for PSMAf is rather controlled, suggesting that the functionalization of PSMA with amino acids would have an effect on the wettability properties of the mat. Among the PSMAf mats, a higher absorption capacity is observed for PSMA@Phe with respect to PSMA@Tyr and PSMA@Gln, with a slightly increasing trend as the days of immersion elapse. The high variability in the water absorption capacity of PSMA is attributed to the mass loss observed during the immersion time, which starts with a visually detectable swelling process of the fibers, followed by a partial solubilization of the polymer. Additionally, the porosity percentages and pore diameter of each mat were measured in order to evaluate a possible effect of the morphological characteristics on the water absorption capacity of the samples. The porosity percentages obtained for PSMA, PSMA@Phe, PSMA@Gln and PSMA@Tyr were 53.5, 53.9, 50.1 and 48.2%, while the average pore diameters were 491.4, 506.6, 510.7 and 535.9 nm, respectively. Although the percentages of porosity and average diameter vary slightly among the mats, it is possible to observe a relationship between both parameters and the water uptake capacity of the mats. More specifically, the mats with higher porosity have higher water absorption capacity, which would suggest that the wettability properties of these polymeric systems are also affected by changes in the morphology of the mats. 

The water absorption capacity of the mats containing 0.5 and 2.0 wt.% AgNPs (see Appendix A) also show a high variability for PSMA associated with swelling and loss of physical integrity of the mat. Among PSMAf, PSMA@Phe presents the highest absorption percentages followed by PSMA@Tyr and PSMA@Gln with a slight increasing trend over time. No correlation between the water absorption capacity and the concentration of AgNPs is observed. 

On the other hand, among the mats containing 5.0 and 20 wt.% Cal (see Appendix A), PSMA also shows variability in its water absorption capacity. In the case of PSMAf, the absorption capacity presents a slightly increasing trend over time, being higher for PSMA@Phe and PSMA@Tyr in the absence of Cal, while for PSMA@Gln, the presence of 5.0 wt.% Cal improves its adsorption capacity. In all cases, the presence of 20 wt.% Cal decreases the water absorption capacity associated with the apolar character of the extract.

The contact angle values shown in Figure 7 show that the functionalization with Gln generates more hydrophilic mats, followed by the functionalization with Tyr and Phe, with the latter being more hydrophobic. However, the results obtained in the water absorption capacity show that PSMA@Phe presents a higher absorption percentage. This behavior could be due to a change in the conformation of the polymeric chains after longer immersion times in water. This conformational change would involve a re-orientation of the aromatic rings present in the Phe residues towards the interior of the fiber, thus establishing π–π interactions between the phenyl group rings. In turn, this conformational change would generate a greater exposure of polar groups such as carboxylic acid towards the external medium of the polymeric fiber, thus generating a more hydrophilic PSMA@Phe mat.

Then, PSMA@Tyr and PSMA@Gln show similar water absorption percentages. In the case of PSMA@Tyr, the absorption capacity would be related to the presence of -OH groups in the Tyr residues, while in PSMA@Gln, to the presence of -NH_2_ groups in the Gln residues. The exposure of these groups to the external environment would be in balance with aromatic rings of the styrene group of the copolymer main chain, exhibiting a hydrophilic/hydrophobic balance.

The conformational changes of functionalized copolymers with amino acids have been previously described in other works, where the wettability properties of these polymeric systems also depend on the hydrophilic/hydrophobic balance of the groups exposed to the external environment. For example, it has been observed that methionine-functionalized PSMA presents -COOH groups exposed to the external environment, resulting in a surface with a less hydrophobic character, despite the more positive methionine hydropathic index, while PSMA-Gln can take the form of a loop exposing the phenyl group of the main polymer chain to the external environment, resulting in hydrophobic domains on the surface [58,59].

### 3.4. Release of Silver Ions from the Mats

Figure 10 shows the silver ion release plots from PSMA and PSMAf containing 0.5 and 2.0 wt.% AgNPs. Among all the mats, PSMA with 2.0 wt.% AgNPs releases the highest amount of silver ions, yielding quantities up to 3.3 ppm. On the other hand, PSMAf mats with 2.0 wt.% AgNPs release concentrations up to 20 times lower than PSMA, where PSMA@Phe releases the lowest amount, reaching values of 0.14 ppm. This result demonstrates that the change of wettability would also influence the release capacity of active agents, controlling, in this case, the Ag^+^ release significantly. 

In order to define the Ag^+^ release behavior, zero-order, first-order, Higuchi, and Korsmeyer–Peppas release models were fitted to the data. The fitting parameters of each model are shown in Table 1. It is observed that in all cases, as the Ag^+^ ion concentration increases from 0.5% to 2.0 wt.%, the Ag^+^ ion release rate also increases. For PSMA, the release rate is on average linear in time for both concentrations, which is reflected through a correlation coefficient (R^2^) close to 1 for a fit based on the zero-order model with zero initial concentration in the dissolution medium. The fit with the first-order model, for both concentrations, is practically indistinguishable from the zero-order model fit because the dissolution process is far from the saturation regime. Consistently, C_1_ K_1_ ≈ C_0_ for both concentrations. Finally, note that although the fit of the PSMA 2.0 wt.% data with the Korsmeyer–Peppas model yields an even higher R^2^, the exponent n = 1.34 corresponds to a superdiffusive process, which would require an active release mechanism. In the case of PSMA@Gln, the release rates for both concentrations are well fitted by the Higuchi model, with an R^2^ slightly below 1. The Korsmeyer–Peppas model also fits well in both cases, with an R^2^ close to 1, although the exponent n for PSMA@Gln 2.0 wt.% falls below the accepted values reported in the literature for slab geometries [51]. On the other hand, in PSMA@Phe 0.5 wt.%, a very steep initial release rate is observed up to t = 0.5 days, followed by a lower and more average constant rate. The same behavior is observed, although to a lesser degree, for PSMA@Phe 2.0 wt.%. The best fit is given by the Korsmeyer–Peppas model for both concentrations, although in the case of PSMA@Phe 0.5 wt.%, the exponent n is unrealistic. In contrast, for PSMA@Phe 2.0 wt.%, n has a value corresponding to non-Fickian transport. Considering only the data from t = 0.5 days (the initial value at t = 0 is discarded) for both concentrations, and assuming an initial concentration C_0_ of Ag^+^ in the medium, the best fit is given by the zero-order model (fitting parameters in parentheses) for both concentrations. Finally, in the case of PSMA@Tyr 0.5 wt.%, the Higuchi model fits the data very well. However, in PSMA@Tyr 2.0 wt.%, a trend change is seen for times longer than 5 days. The model that best represents the data in this case is the Korsmeyer–Peppas model, although with an exponent n = 1.42 (superdiffusive). By considering the first 5 days of release for PSMA@Tyr 2.0 wt.%, the best fit is given by the first-order model (fitting parameters in parentheses). The Korsmeyer–Peppas model also fits well, with an exponent n typical of an anomalous transport process.

### 3.5. Release of C. officinalis from the Mats

Figure 11 shows the Cal release plots from PSMA and PSMAf containing 5 and 20 wt.% Cal. Among the PSMAf curves, it is observed that PSMA@Tyr mats release the highest amount of Cal, followed by PSMA@Gln and finally PSMA@Phe. If we compare the release behavior with the water absorption capacity, an inverse behavior is observed where the percentages of Cal release are higher for those mats that present a lower interaction with water. This behavior would be related to the apolar character of Cal, which would inhibit the interaction of the mats with water. In general, the release of active agents will depend on the wettability of the fibers in the mat. However, when the active agent to be released in an aqueous medium is a compound of apolar nature, its diffusion will be limited and will not depend exclusively on the capacity of the water molecules to penetrate between the fibers and polymeric chains, so wettability will not be the factor that governs diffusion. Based on this, a high variability in the results is expected as indeed is shown in the graphs in Figure 11. On the other hand, interactions could also occur between the Cal and the functional groups of the copolymer that would retain it, preventing its release. This could explain the lower release on day 7 for PSMA@Tyr.

### 3.6. Antibacterial Activity of Polymer Mats

The antibacterial activity of the mats was evaluated by bacterial colony count assay on a Mueller–Hinton agar medium. Figure 12 shows the agar plates containing the bacterial colonies of *S. aureus* ATCC 25923 [SA] along with the graphs of percent bacterial growth for (A) PSMA, (B) PSMA@Gln, (C) PSMA@Tyr and (D) PSMA@Phe. In all cases, the bacterial culture was incubated for 1 h in the presence of the 1-day and 7-day release solution (Ag^+^ in deionized water). The figure shows that the PSMA mat inhibited 99.9% of SA growth, while among the PSMAf mats, only a slight growth inhibition and, in some cases, a proliferative effect were observed. Among the functionalized mats, PSMA@Gln exhibited the highest percentage inhibition of SA incubated in the 1-day release solution, reaching 56.2% inhibition, followed by PSMA@Tyr with 10%, while a proliferative effect was observed in PSMA@Phe. On the other hand, SA incubated in the 7-day release solutions showed a slight proliferative effect for PSMA@Tyr and a more significant one for PSMA@Phe, while in PSMA@Gln there was no inhibition or proliferative effect.

Appendix A shows the agar plates containing the bacterial colonies of methicillin-resistant *S. aureus* ATCC 33592 [MRSA] along with the graphs of percent bacterial growth for (A) PSMA, (B) PSMA@Gln, (C) PSMA@Tyr and (D) PSMA@Phe. As shown in Appendix A PSMA inhibited by 99.9% the bacterial growth of MRSA incubated in the 1-day and 7-day release solutions, while for PSMA@Gln containing 0.5 wt.% AgNPs, a percentage inhibition close to 30% is observed, and a proliferative effect for PSMA@Tyr is observed in the 1-day release solutions. For MRSA incubated in 7-day release solutions, a proliferative effect close to 7% for PSMA@Gln and between 2 and 20% for PSMA@Tyr is again observed. The low inhibition percentages and in some cases the proliferative effect of the functionalized PSMA mats could be related to three characteristics: first, the difference in wettability of the mats; second, a possible chelating effect associated with the structure of the amino acid residue present in the copolymer chain; and third, the presence of free amino acid in the mat released during the immersion time in the aqueous medium. In the first case, it is expected that in hydrophobic mats, such as PSMA@Tyr and PSMA@Phe, the diffusion of water molecules into the fiber will be slower and so will be the subsequent diffusion of Ag^+^ ion. Thus, a low concentration of Ag^+^ released would not be sufficient to carry out significant bacterial inhibition. This result is further confirmed by the Ag^+^ release curves measured by ICP-MS, where the concentrations for PSMA@Tyr and PSMA@Phe were up to three times lower than PSMA. In the second case, it is known that amino acids such as glutamine can have a chelating effect on some metal ions. Glutamine has been used as a treatment against biochemical and histopathological alterations induced by lead in rat livers [60]. In our case, it is possible that the release of Ag^+^ from the fibers could have been prevented by the chelating effect associated with the -NH_2_ group present in Gln. In the case of Tyr and Phe, the chelating effect could be related to cation–π interactions between the aromatic ring of the amino acid and the Ag^+^ ion. Regarding the third case, previous studies have shown that amino acids such as glutamine and tyrosine promote bacterial growth of both Gram-positive and Gram-negative bacteria and have been used as a carbon or nitrogen source or as a suppressive agent of some biocides [60,61,62,63,64]. Therefore, we believe that the presence of free amino acids would be promoting bacterial growth. 

### 3.7. Cytotoxicity of Polymer Mats

Figure 13 shows the cytotoxic effect of PSMA and PSMAf containing different concentrations of Cal on a murine NIH/3T3 fibroblast cell line. In the case of PSMA and PSMA@Tyr, the highest viability percentages are observed, with a slight proliferative effect for the mats containing AgNPs, while PSMA@Phe and PSMA@Gln show slightly lower percentages in the absence of a proliferative effect. According to the Cal release results (Figure 11), there could be a correlation between the mats that present higher release (PSMA@Tyr) and those that showed an increase in cell viability. In general, there is no significant variation in the percentage of cell viability of NIH/3T3 fibroblasts, except for the PSMA 2 wt.% AgNPs mats that generated an increase in cell viability. The latter could be due to some stimulus generated by the presence of AgNPs related to the survival of fibroblasts that have been subjected to environments containing reactive oxygen species generated by AgNPs [65].

### 3.8. Hemolytic Activity of Polymer Mats

Figure 14 shows the hemolysis percentages obtained for PSMA and PSMAf. PSMA shows the lowest hemolysis percentages, with values between 4% and 6%, followed by PSMA@Tyr and PSMA@Gln, reaching maximum hemolysis percentages of 9.90% for the latter. The highest percentages are observed for PSMA@Phe, exceeding the percentage of the Triton X-100 positive control. The hemolytic effect of PSMA@Phe could be due to a surface effect of the mat on the erythrocyte membranes, where certain functional groups on the surface of the fibers interact with the erythrocyte membrane, generating instability and subsequent rupture. This behavior is similar to that caused by certain amphipathic drugs, which generate instability and cause subsequent cell rupture [66]. This result is consistent with the water absorption behavior (Figure 9), where a possible conformational change of the polymeric chains in PSMA@Phe was discussed, which would involve a re-orientation of the aromatic rings towards the interior of the fiber and a greater exposure of the polar groups such as carboxylic acids to the external environment of the polymeric fiber. It is possible that the exposure of these groups and their interaction with the cell membrane of the erythrocytes are responsible for the hemolytic effect. This same phenomenon could explain the slight hemolytic response of the PSMA@Gln and PSMA@Tyr mats, but with a lower exposure and participation of these groups in cell lysis.

## 4. Conclusions

Polymer mats were prepared from PSMA and PSMA functionalized with Gln, Phe and Tyr. The results of contact angle and water absorption capacity measurements showed that the copolymer functionalization had an effect on the wettability properties of the mats. The highest contact angle values were obtained for PSMA@Phe followed by PSMA@Tyr and PSMA@Gln, which were conditioned with the water index values of each amino acid. However, the water absorption capacity differs from this behavior, which is attributed to conformational changes of the polymer chains after prolonged immersion times. The differences observed in the wettability of each type of mat, as well as in their structural characteristics, were mainly reflected in the Ag^+^ release behavior, where a significant release was observed for PSMA and a more controlled release for all functionalized PSMA mats. The results of antibacterial capacity against *S. aureus* ATCC 25923 and methicillin-resistant *S. aureus* ATCC 33592 revealed that PSMA release solutions containing only 0.5 wt.% of AgNPs inhibited 99.99% of the growth of both strains after 1 h of contact time with the release solution, while PSMA@Gln, PSMA@Tyr and PSMA@Phe mats presented a slight effect, or in some cases a proliferative effect. On the other hand, cell viability results of NIH/3T3 fibroblasts showed a slight increase in the quantity of active agents released from the mats, specifically for those containing 2.0% AgNPs. Additionally, hemolysis results revealed that PSMA@Tyr and PSMA@Gln mats presented hemolysis percentages lower than 10%, while PSMA@Phe presented a high degree of hemolytic over 100%, which could be attributed to the structural conformation adopted by PSMA@Phe on the fiber surface. Finally, the variable wettability of the mats, as well as in their structural characteristics, showed an effect on the modulated release of the active agents and on their bioactivity, where the rapid release of Ag^+^ from the PSMA mats could see potential use in a dressing for a wound susceptible to infection, while PSMA@Tyr could be more suitable for a slow healing wound. Thus, we conclude that the use of PSMA copolymers can be a functionalization platform capable of modulating the release of active agents to suit the treatment of wounds of different characteristics. 

## Figures and Tables

**Figure 1 pharmaceutics-15-01659-f001:**
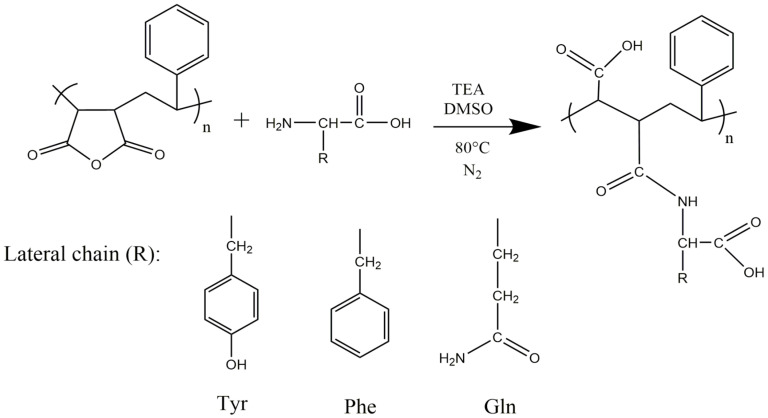
Schematic of the functionalization reaction of PSMA with different amino acids.

**Figure 2 pharmaceutics-15-01659-f002:**
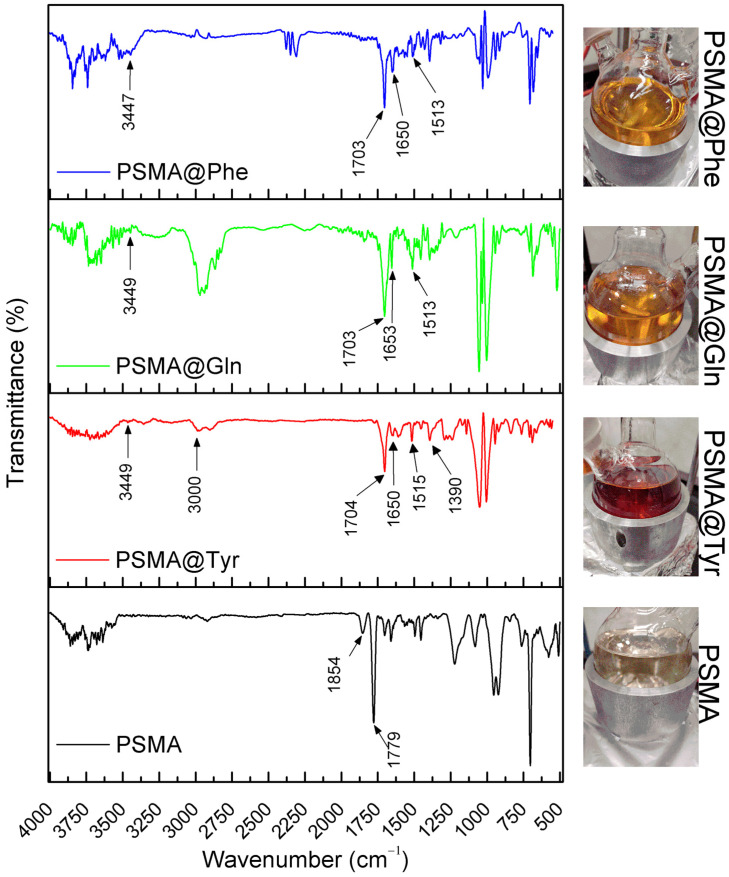
ATR-FTIR spectra of mats of PSMA and PSMAf.

**Figure 3 pharmaceutics-15-01659-f003:**
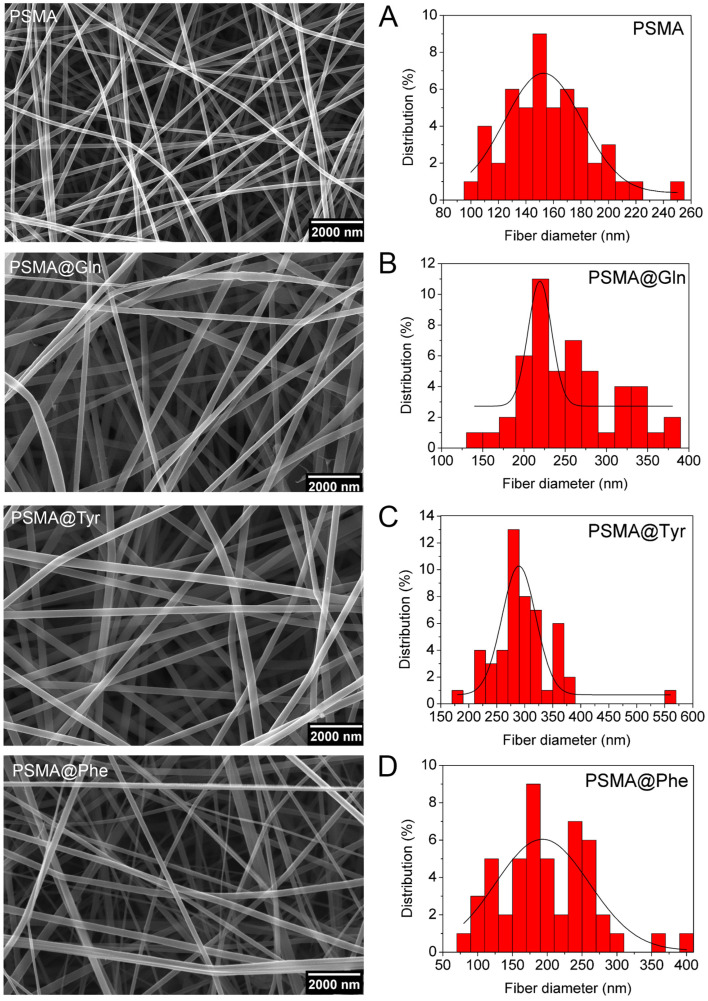
SEM images of mats of (**A**) PSMA, (**B**) PSMA@Gln, (**C**) PSMA@Tyr and (**D**) PSMA@Phe. Magnification 30,000×.

**Figure 4 pharmaceutics-15-01659-f004:**
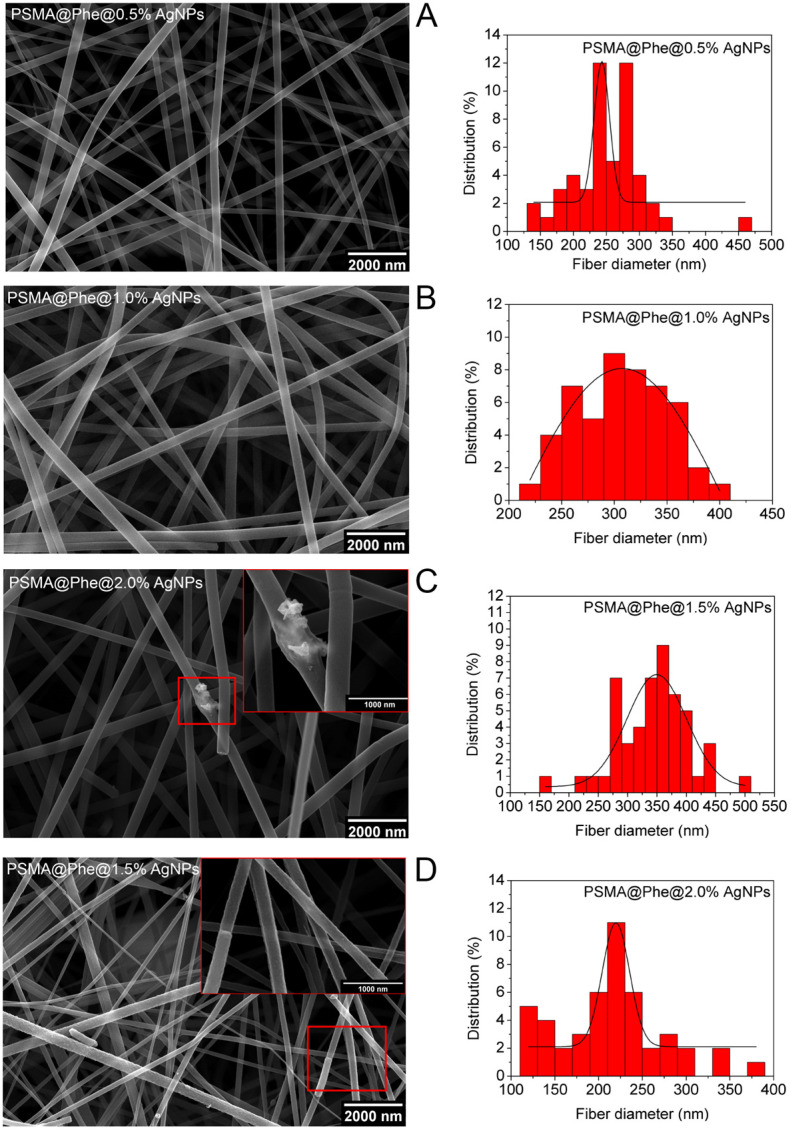
SEM images PSMA@Phe with different concentrations of AgNPs, (**A**) 0.5, (**B**) 1.0, (**C**) 1.5 and (**D**) 2.0 wt.% Inset in image (**C**) and (**D**) corresponds to higher magnification area. Magnification 30,000×.

**Figure 5 pharmaceutics-15-01659-f005:**
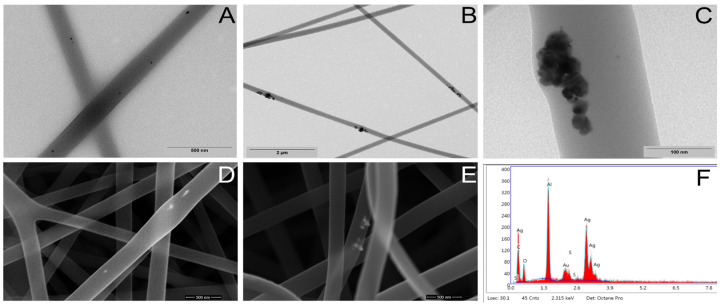
(**A**) TEM image of PSMA@Phe fibers containing 0.5 wt.% AgNPs, (**B**) TEM image of PSMA@Phe fibers containing 2.0 wt.% AgNPs, (**C**) 100,000× magnification image of PSMA@Phe containing 2.0 wt.% AgNPs, (**D**) FESEM image of PSMA@Phe mats containing 1.0 wt.% AgNPs, (**E**) FESEM image of PSMA@Phe mats containing 1.5 wt.% AgNPs and (**F**) X-ray energy dispersive spectrum (EDS) of image (**E**).

**Figure 6 pharmaceutics-15-01659-f006:**
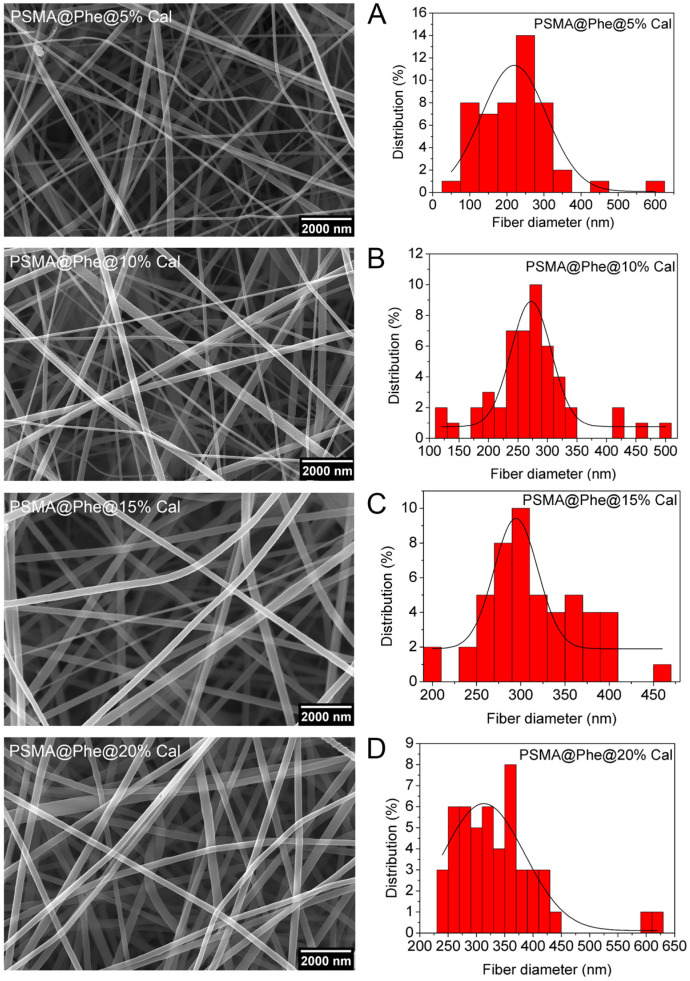
SEM images PSMA@Phe with different concentrations of *C. officinalis*, (**A**) 5.0, (**B**) 10, (**C**) 15 and (**D**) 20 wt.%. Magnification 30,000×.

**Figure 7 pharmaceutics-15-01659-f007:**
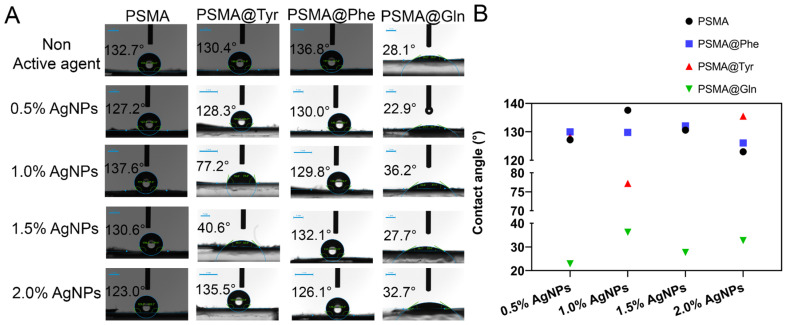
(**A**) Contact angles of PSMA and PSMAf in the absence and presence of 0.5, 1.0, 1.5 and 2.0 wt.% AgNPs (scale bar: 1 mm), (**B**) Plot of the contact angles of PSMA and PSMAf.

**Figure 8 pharmaceutics-15-01659-f008:**
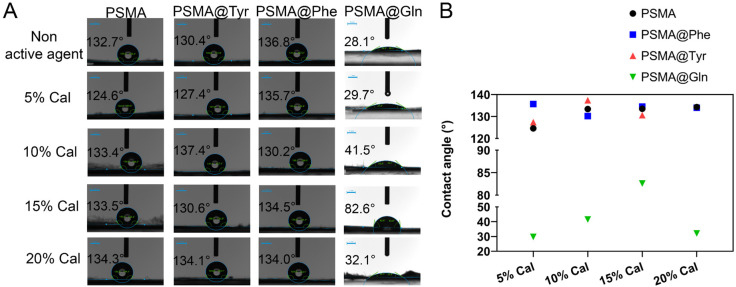
(**A**) Contact angles of PSMA and PSMAf in the absence and presence of 5, 10, 15 and 20 wt.% of *C. officinalis* (scale bar: 1 mm), (**B**) Graph of the contact angles of PSMA and PSMAf.

**Figure 9 pharmaceutics-15-01659-f009:**
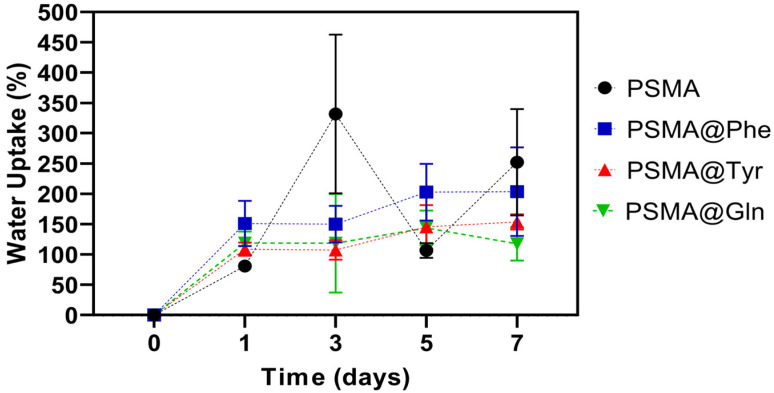
Water absorption capacity of PSMA and PSMAf over time.

**Figure 10 pharmaceutics-15-01659-f010:**
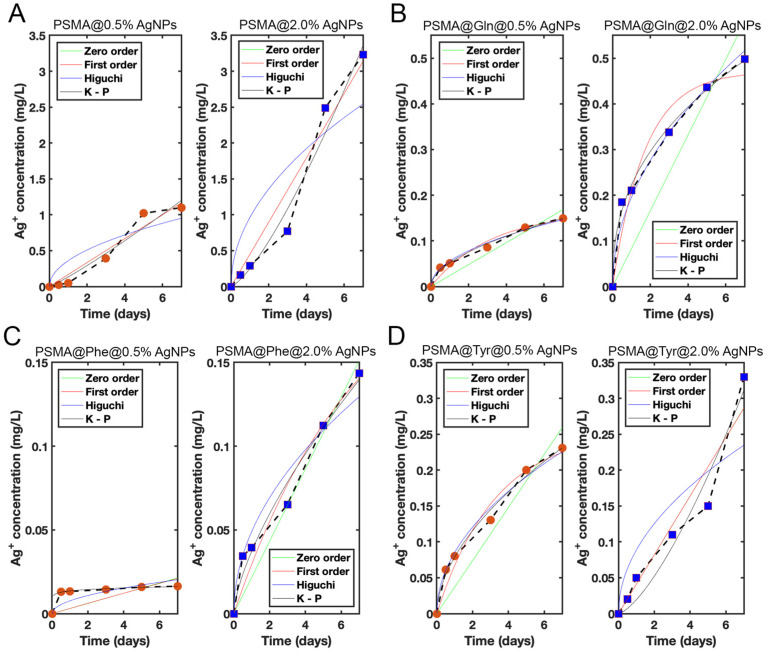
Ag^+^ release curves from mats containing 0.5 wt.% (red circle) and 2.0 wt.% (blue square) concentrations of AgNPs: (**A**) PSMA, (**B**) PSMA@Gln, (**C**) PSMA@Phe and (**D**) PSMA@Tyr.

**Figure 11 pharmaceutics-15-01659-f011:**
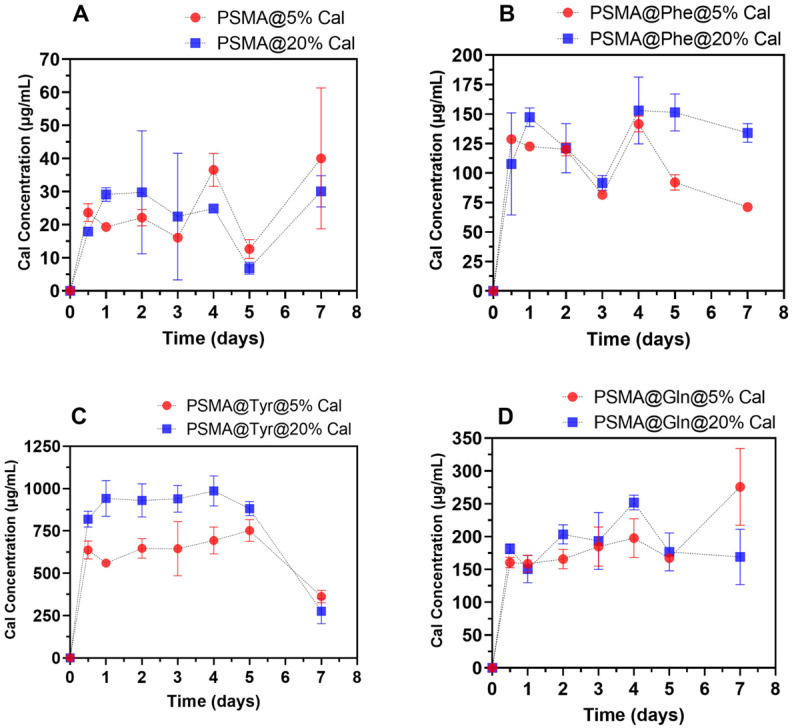
Release of *C. officinalis* from mats containing 5 and 20 wt.% concentrations of Cal. (**A**) PSMA, (**B**) PSMA@Phe, (**C**) PSMA@Tyr and (**D**) PSMA@Gln.

**Figure 12 pharmaceutics-15-01659-f012:**
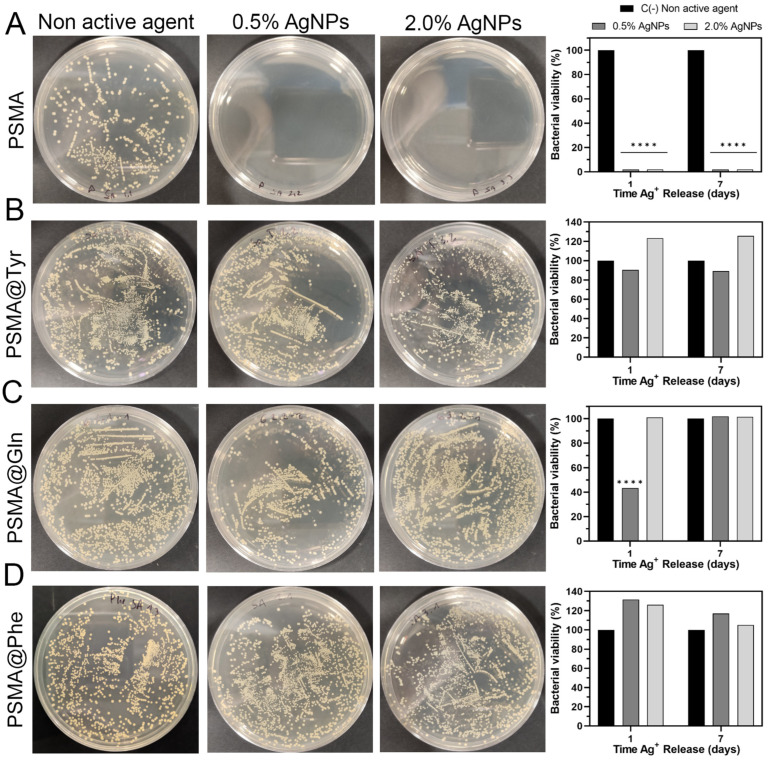
Bacterial viability of *S. aureus* ATCC 25923 incubated in release solutions of, (**A**) PSMA, (**B**) PSMA@Gln, (**C**) PSMA@Tyr and (**D**) PSMA@Phe. Data were analyzed using Dunnett’s multiple comparisons test; **** *p* < 0.0001 compared to the negative control group (non active agent).

**Figure 13 pharmaceutics-15-01659-f013:**
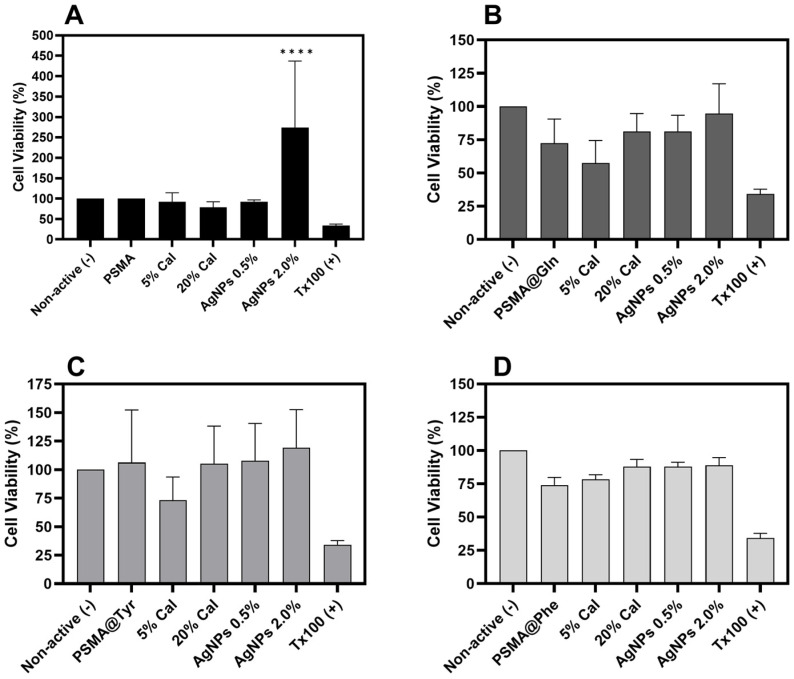
Percentage cell viability of NIH/3T3 fibroblasts versus PSMA and PSMAf mats with different concentrations of active agents. (**A**) PSMA, (**B**) PSMA@Gln, (**C**) PSMA@Tyr and (**D**) PSMA@Phe. Data were analyzed using Dunnett’s multiple comparisons test; **** *p* < 0.0001 compared to non-active.

**Figure 14 pharmaceutics-15-01659-f014:**
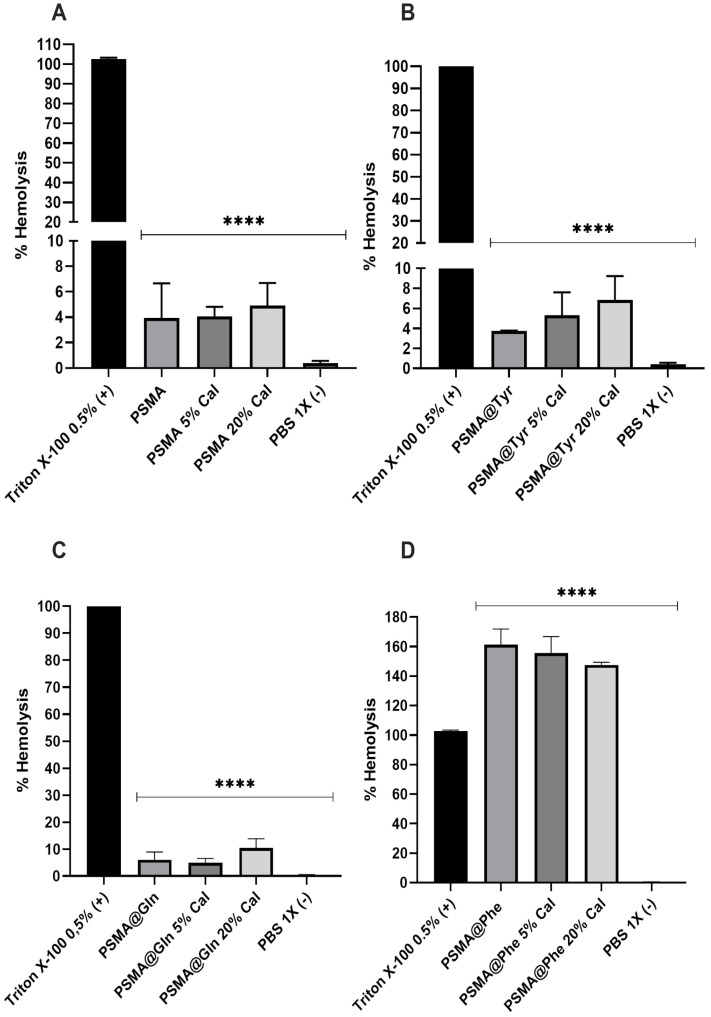
Percent hemolysis of polymeric mats (PSMA and PSMAf) with different concentrations of active agents. (**A**) PSMA, (**B**) PSMA@Gln, (**C**) PSMA@Tyr and (**D**) PSMA@Phe. Data were analyzed using Dunnett’s multiple comparisons test; **** *p* < 0.0001 compared to the positive control group (Triton X-100).

**Table 1 pharmaceutics-15-01659-t001:** Fit parameters for the different mathematical models.

Model	Param.	PSMA		PSMA-Gln		PSMA-Phe		PSMA-Tyr	
		0.5%	2.0%	0.5%	2.0%	0.5%	2.0%	0.5%	2.0%
Zero-order	C_0_	-	-	-	-	−(0.0128)	−(0.0219)	-	-
	K_0_	0.167	0.448	0.0240	0.0829	0.00305 (0.000530)	0.0216 (0.0173)	0.0370	0.0410 (0.0323)
	R^2^	0.947	0.954	0.861	0.690	−0.570 (0.979)	0.933 (0.988)	0.868	0.931 (0.960)
First-order	C_1_	134	3003	0.164	0.471	0.266	0.224	0.255	33.1 (0.210)
	K_1_	0.00125	0.000149	0.312	0.585	0.0117	0.141	0.305	0.00124 (0.249)
	R^2^	0.947	0.954	0.963	0.951	−0.538	0.956	0.965	0.930 (0.998)
Higuchi	K_H_	0.360	0.961	0.0554	0.195	0.00764	0.0489	0.0853	0.0887 (0.0621)
	R^2^	0.790	0.774	0.989	0.984	0.412	0.952	0.989	0.774 (0.946)
Korsmeyer–Peppas	n	1.13	1.34	0.544	0.412	0.0882	0.679	0.552	1.42 (0.737)
	K_KP_	0.134	0.250	0.052	0.222	0.0136	0.0372	0.0789	0.0194 (0.0467)
	R^2^	0.941	0.969	0.984	0.992	0.925	0.964	0.985	0.940 (0.989)

## Data Availability

The data presented in this study are available on request from the cor-responding author.

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
