# Peer review of "Wettability of Amino Acid-Functionalized PSMA Electrospun Fibers for the Modulated Release of Active Agents and Its Effect on Their Bioactivity"

_pharmaceutics, 2023, doi:10.3390/pharmaceutics15061659_

Round 1

Reviewer 1 Report

Manuscript is well written with appropriate design of experiments. However, discussion is really poor, most of the conclusions are not supported by the publications.

1)    Page 3 line 114, the term “cell regeneration” it not applicable here; cell proliferation, migration etc. might be better.

2)    Last paragraph of introduction should provide brief information about obtained results, please expand this part.

3)    Please provide the unit of molecular weight of polymer.

4)    Please provide the details of ATR-FTIR measurement.

5)    Please provide the humidity during the electrospinning process.

6)    In the reviewer opinion, in the cytotoxicity assay, untreated cells should be added as a control.

7)    Page 9 line 333, the term “characterization” might be better instead of “elaboration”.

8)    Please provide bigger scale bars, they are barely visible. Consider 100 nm or 500 nm scale bar instead of 1000 nm. How the samples for SEM observation were prepared?

9)    How many fibers were calculated to obtain fiber diameter, please provide average fiber diameter and standard deviation. 

10) Page 10 lines 344-346, please confirm this statement within the citation.

11) Contact angle results obtained for mats with AgNPs did not follow any trend, may it indicate not homogenous distribution of AgNPs, please comment it. How the contact angle correlates with fibers size.

12) Please discuss results presented in 3.3.2. with other works.

13) Why authors choose indirect microbiology test? Usually, it this kind of assays, electrospun membranes are placed on the bacterial colony and growth inhibition zone is measured.

14) Page 24 lines 658-659, does the cell viability really increase with the amount of released active agents? According to the statistic, I do not see difference in cell proliferation. 

15) Conclusion needs some correlation with the introduction, so how those mats can be used in wound treatment and why they are better than existing ones.

Author Response

Reviewer 1

1)    Page 3 line 114, the term “cell regeneration” it not applicable here; cell proliferation, migration etc. might be better.

We thank the reviewer for his/her comment. We have changed the term “cell regeneration” to “cell proliferation”

2)    Last paragraph of introduction should provide brief information about obtained results, please expand this part.

We thank the reviewer for his/her comment. We have included brief information about the results in the last paragraph of the introduction.

“The results of contact angle and water absorption capacity measurements showed that the copolymer functionalization had an effect on the wettability properties of the mats, where he differences observed in the wettability, as well as in their structural characteristics, were mainly reflected in the Ag+ release behavior and antibacterial response.”

3)    Please provide the unit of molecular weight of polymer.

We thank the reviewer for his/her comment. We have included the unit of molecular weight in the manuscript.

4)    Please provide the details of ATR-FTIR measurement.

We thank the reviewer for his/her comment. We have included more details of ATR-FTIR measurement.

“The spectra were performed at 24°C with a relative humidity of 40%. The spectra were collected in the mats or reaction product by averaging 10 scans at 2 cm−1 resolution.”

5)    Please provide the humidity during the electrospinning process.

We thank the reviewer for his/her comment. We have included information about relative humidity (%) during the electrospinning process.

“…relative humidity of 43% and a voltage of 27 kV.”

6)    In the reviewer opinion, in the cytotoxicity assay, untreated cells should be added as a control.

We agree with the opinion of the reviewer, however, our objective is mainly focused on the effect of functionalization on the biological response, so we consider it more ad hoc to use cells treated with PSMA as a negative control.

7)    Page 9 line 333, the term “characterization” might be better instead of “elaboration”.

We thank the reviewer for his/her comment. We have changed the term "elaboration" to "characterization".

8)    Please provide bigger scale bars, they are barely visible. Consider 100 nm or 500 nm scale bar instead of 1000 nm. How the samples for SEM observation were prepared?

We thank the reviewer for his/her comment. We have provided bigger scale bars on the SEM images. In addition, we have included a brief text on sample preparation.

“The mats obtained were characterized by field emission scanning electron microscopy (FESEM; FEI Quanta FEG250) operated at 15kV”

“The samples were coated with thin films of 5 nm of gold, prior to being observed.”

9)    How many fibers were calculated to obtain fiber diameter, please provide average fiber diameter and standard deviation.

50 fibers were used for the calculation of the average diameter. The standard deviation in each case was included in the manuscript.

“According to the histograms, fibers made from PSMA@Tyr have the largest average diameter (3.0 ± 0.6) ´ 102 nm, followed by PSMA@Gln (2.6 ± 0.5) ´ 102 nm, and PSMA@Phe (2.0 ± 0.6) ´ 102 nm. The fibers obtained from PSMA show a smaller diameter (1.6 ± 0.3) ´ 102 nm compared to all PSMAf mats.”

10) Page 10 lines 344-346, please confirm this statement within the citation.

We thank the reviewer for his/her comment. We have included a reference confirming this statement.

“The increasing diameter fiber due to steric hindrance between the polymer chains has also been described by Torricelli et al.”

11) Contact angle results obtained for mats with AgNPs did not follow any trend, may it indicate not homogenous distribution of AgNPs, please comment it. How the contact angle correlates with fibers size.

We thank the reviewer for his/her comment. Indeed, the distribution of the AgNPs in the mesh is not homogeneous. This was confirmed by the TEM and FESEM images that revealed the presence of non-periodic clusters within the fibers, which could explain the absence of a trend in the measured values of the contact angle when changing the concentration of AgNPs. On the other hand, although the topography of the meshes could have an effect on the value of the contact angle, in our case no correlation was observed with the fiber diameter.

12) Please discuss results presented in 3.3.2. with other works.

We have included a paragraph citing previous works in which the wettability properties of polymeric systems depend on the hydrophilic/hydrophobic balance of the chemical groups exposed to the external environment.

“The conformational changes of functionalized copolymers with amino acids have been previously described in other works, where the wettability properties of these polymeric systems also depend on the hydrophilic/hydrophobic balance of the groups exposed to the external environment. For example, it has been observed that methionine-functionalized PSMA presents –COOH groups exposed to the external environment, resulting in a surface with a less hydrophobic character, despite the more positive methionine hydropathic index, while PSMA-Gln can take the form of a loop exposing the phenyl group of the main polymer chain to the external environment, resulting in hydrophobic domains on the surface. [58,59]”

13) Why authors choose indirect microbiology test? Usually, it this kind of assays, electrospun membranes are placed on the bacterial colony and growth inhibition zone is measured.

We thank the reviewer for his/her comment. We, however, consider that the use of release media for antibacterial assays is more suitable for the focus of this work, since in this way we can directly relate the behavior of Ag+ release over time with the antibacterial capacity of the mats.

14) Page 24 lines 658-659, does the cell viability really increase with the amount of released active agents? According to the statistic, I do not see difference in cell proliferation.

We thank the reviewer for his/her comment. We consider that his/her observation is very pertinent. We included a brief paragraph correcting this conclusion.

“On the other hand, cell viability results of NIH/3T3 fibroblasts showed a slight increase with the quantity of active agents released from the mats, specifically for those containing 2.0% AgNPs.”

15) Conclusion needs some correlation with the introduction, so how those mats can be used in wound treatment and why they are better than existing ones.

We thank the reviewer for his/her comment. We have included in the conclusion a paragraph that discusses the applications of our results to wound dressings, in connection with what is said in the introduction.

“… where the rapid release of Ag+ from the PSMA mats could be a potential dressing for a wound susceptible to infection, while PSMA@Tyr could be more suitable for a slow healing wound. Thus, we can conclude that the use of PSMA copolymers can be a functionalization platform capable of modulating the release of active agents to suit the treatment of wounds of different characteristics.” 

Reviewer 2 Report

Amino acid functionalized PSMA nanofibers were fabricated for modulated release of active agents like silver nanoparticles and Calendula officinalis and its bioactivity also studied. This manuscript can be accepted after minor revision.

1. FTIR spectrum of nanofibers loaded with AgNPs and Cal should be included and nature of interaction between functionalized PSMA and the bioactive materials should be discussed.

 2. I wonder the author reported 99.9% bacterial inhibition achieved in 1 h incubation. This statement is completely wrong. The minimum incubation time should be 8 h – 24 h.

Author Response

Reviewer 2

Amino acid functionalized PSMA nanofibers were fabricated for modulated release of active agents like silver nanoparticles and Calendula officinalis and its bioactivity also studied. This manuscript can be accepted after minor revision.

  1. FTIR spectrum of nanofibers loaded with AgNPs and Cal should be included and nature of interaction between functionalized PSMA and the bioactive materials should be discussed.

We thank the reviewer for his/her comment. The FTIR spectra of the nanofibers loaded with AgNPs and Cal were included in the Supplementary Material file. In addition, an analysis on the nature of the interaction between functionalized PSMA with the bioactive agents was included in the Results section.

“Additionally, the ATR-FTIR spectra of the PSMA and PSMAf mats in the absence and presence of the active agents of AgNPs (2.0 wt.%) and Cal (20 wt.%) were also obtained. The spectra are shown in Figures S1, S2, S3 and S4 in Supplementary Material. For PSMA, the signal observed at 1643 cm-1 is attributed to the torsion of the C=C bond, however, a broadened signal is observed for the mats containing active agents, suggesting a possible interaction between the aromatic styrene ring present in the PSMA chain with the AgNPs and Cal. In the case of Cal the interaction would be of the ?-? type between the respective aromatic rings. In the case of PSMA@Phe in the presence of the active agents, a displacement of the signal by ~1000 cm-1 is attributed to the deformation of the C-H bond of a monosubstituted alkene present in the aromatic ring of Phe, indicating a possible interaction of the Phe ring with AgNPs and Cal. On the other hand, for PSMA@Tyr containing active agents the presence of broadened signal at 1700 cm-1 indicates a possible interaction of the carbonyl group of the ring opening with AgNPs and Cal. Finally, PSMA@Gln shows a signal displacement of 764 cm-1suggesting a possible interaction of the amine group of the amino acid residue with AgNPs and Cal.”

  1. I wonder the author reported 99.9% bacterial inhibition achieved in 1 h incubation. This statement is completely wrong. The minimum incubation time should be 8 h – 24 h.

We thank the reviewer for his/her comment. The incubation time of 1 hour refers to the contact time of the bacterial culture with the release solution containing the Ag+ ions. On the other hand, the agar plates were incubated for 24 h at 37°C. To avoid confusion, we replaced the word "incubation" with "contact time" in the manuscript.

Reviewer 3 Report

Dear editor,

The manuscript (pharmaceutics-2399179) reports about the development of bioactive polymeric fibers mats made of poly(styrene-co-maleic anhydride) [PSMA] that have been functionalized with L-glutamine, L-phenylalanine and L-tyrosine with different levels of wettability. The fibers have been subsequently loaded with Ag nanoparticles or Cal. The research idea is interesting, the manuscript reads well, and required characterizations have been carried out. However, there are several major concerns that should be addressed:

1- Page 2, lines 50-58 needs references such as: Acta biomaterialia 107, 25-49, 2020, Materials Science and Engineering: C 116, 111248, 2020, and Materials Science and Engineering: C 123, 111965, 2021.

2- Given the presence of Cl ions in biological liquids, how Ag+ ions do not precipitate as AgCl and still show antibacterial activity?

3- Page 4, section 2.4; addition of Cal to DMF and DCM does not negatively affect its structure and properties?

4- FESEM; how much was the electron accelerating voltage and if the samples were sputter coated?

5- How the nanofiber samples were sterilized for in vitro tests?

6- How reliable would be the cell test data after only 24 hr incubation? If possible, extend the duration and report the cell viability data over a longer cell culture time with several intervals.

7- Page 9, lines 338-340; the nanofiber diameters need to have standard deviation.

8- Considering the entrapment of Ag NPs inside the nanofibers, how they release Ag+ ions, thereby killing the bacteria?

9- Page 15, water uptake capacity; why the authors have not measured the wound exudate uptake capacity instead of water uptake capacity?

10- In addition to wettability and surface chemistry, water uptake capacity can depend on the porosity of the nanofiber mats. I suggest the author to quantify the porosity of the nanofiber mats to better clarify the obtained results.

11- Can the released Ag+ ions be chelated by the functional groups of the amino acids and polymer and re-form Ag NPs on the nanofibers surface?

12- Figure 11a&b; what is the reason for inconsistent release of Cal on the 5th and 3rd day, respectively?

13- Page 21, line 609; how can Ag NPs generate ROS?

Author Response

Reviewer 3

The manuscript (pharmaceutics-2399179) reports about the development of bioactive polymeric fibers mats made of poly(styrene-co-maleic anhydride) [PSMA] that have been functionalized with L-glutamine, L-phenylalanine and L-tyrosine with different levels of wettability. The fibers have been subsequently loaded with Ag nanoparticles or Cal. The research idea is interesting, the manuscript reads well, and required characterizations have been carried out. However, there are several major concerns that should be addressed:

1- Page 2, lines 50-58 needs references such as: Acta biomaterialia 107, 25-49, 2020, Materials Science and Engineering: C 116, 111248, 2020, and Materials Science and Engineering: C 123, 111965, 2021.

We thank the reviewer for his/her comment. The references, Acta biomaterialia 107, 25-49, 2020 and Materials Science and Engineering: C 116, 111248, 2020, and Materials Science and Engineering: C 123, 111965, 2021 were included in the introduction section.

2- Given the presence of Cl ions in biological liquids, how Ag+ ions do not precipitate as AgCl and still show antibacterial activity?

The amount of Ag+ ions obtained from the releases and used in the antibacterial tests, that is, considering the dilutions, are lower than the concentrations calculated from the value of the AgCl solubility product (Kps =1.8 x 10-10). Therefore, such concentrations would not cause the formation of an AgCl precipitate.

3- Page 4, section 2.4; addition of Cal to DMF and DCM does not negatively affect its structure and properties?

We thank the reviewer for his/her comment. Previously, we have obtained FTIR spectra of Cal and Cal solutions in DMF/DCM, where the signals obtained indicate that the structure of this molecule is not affected.

4- FESEM; how much was the electron accelerating voltage and if the samples were sputter coated?

We thank the reviewer for his/her comment. The accelerating voltage used in the FESEM was 15 kV. The value was incorporated into the manuscript.

5- How the nanofiber samples were sterilized for in vitro tests?

The nanofiber samples were sterilized for in vitro tests using a standard UV irradiation method. Working in a laminar flow hood (Purifier Logic+ Class II, Type B2 Biosafety Cabinet, Labconco®) under sterile conditions, materials were cut and carefully placed in a 96 well plate. Nanofiber samples were UV irradiated for 45 min. Well plates with samples were located on the countertop in the laminar flow hood as a UV source, a lamp from the laminar hood was used. Samples were used for testing directly after sterilization. Int. J. Mol. Sci. 2020, 21(21), 8092; https://doi.org/10.3390/ijms21218092

6- How reliable would be the cell test data after only 24 hr incubation? If possible, extend the duration and report the cell viability data over a longer cell culture time with several intervals.

The reliability of cell data after 24-hours of incubation period depends on various factors, experimental conditions, specific cell type, and experimental purpose. In this case, for experimental procedures we considered a standard analysis widely used to estimate in vitro cytotoxicity of nanofibers at 24 h post-culture. Therefore, we analyzed cell mitochondrial activity after 24 hours of cell culture seeding using a procedure according to the manufacturer's protocol (WST-1, Cellpro-Roche, Sigma-Aldrich). In this context, previous studies have shown this approach to estimate short time cell behavior on different materials of nanofibers and microfibers. Daraeinejad Z. and Shabani I. estimated cell relative viability at 24 h through MTT cytotoxicity assay of polyaniline (PANI) coated nanofibers. Fujita, K., Obara S. and Maru J. evaluated cell viability of submicron-diameter carbon fibers (SCFs) after 24 h of NR8383 cells seeding and further analysis with WST-1 assay. Finally, Haider MK., et al. estimate in vitro cytotoxicity of MC3T3 cells cultured at 24 h post seeding on lignin nanoparticles/polycaprolactone nanofibers. We project for the next article a longer-term cell viability profile, based on improved PMSA nanofibers from these initial findings.

7- Page 9, lines 338-340; the nanofiber diameters need to have standard deviation.

We thank the reviewer for his/her comment. The standard deviation of nanofiber diameters were included into the manuscript.

“According to the histograms, fibers made from PSMA@Tyr have the largest average diameter (298.6±60.0 nm) (3.0 ± 0.6) ´ 102 nm, followed by PSMA@Gln (255.9±50.0 nm) (2.6 ± 0.5) ´ 102 nm, and PSMA@Phe (198.6±64.9 nm) (2.0 ± 0.6) ´ 102 nm. The fibers obtained from PSMA show a smaller diameter (156.3±30.9 nm) (1.6 ± 0.3) ´ 102 nm compared to all PSMAf mats.”

8- Considering the entrapment of AgNPs inside the nanofibers, how they release Ag+ ions, thereby killing the bacteria?

Ag+ ions can migrate between the polymeric chains of PSMA or functionalized PSMA, because the fiber is not a compact structure, but rather a thread of polymeric chains with gaps between chains.

9- Page 15, water uptake capacity; why the authors have not measured the wound exudate uptake capacity instead of water uptake capacity?

We thank the reviewer for his/her comments, however, in vivo tests are not contemplated within the objectives of this work, since the emphasis was on the effect of wettability on the release of active agents.

10- In addition to wettability and surface chemistry, water uptake capacity can depend on the porosity of the nanofiber mats. I suggest the author to quantify the porosity of the nanofiber mats to better clarify the obtained results.

We thank the reviewer for his comment. We have quantified the porosity of the mats using the Image J software (NIH, USA) from three different images of the same type of sample. The results were related to the water uptake capacity and incorporated into the manuscript.

“The porosity and average pore diameter of the mats was quantified using Image J software (NIH, USA) using 3 different images for each type of sample according to threshold method. [47,48].”

“Additionally, the porosity percentages and pore diameter of each mat were measured in order to evaluate a possible effect of the morphological characteristics on the water uptake capacity of the samples. The porosity percentages obtained for PSMA, PSMA@Phe, PSMA@Gln and PSMA@Tyr were 53.5, 53.9, 50.1 and 48.2%, while the average pore diameters were 491.4, 506.6, 510.7 and 535.9 nm, respectively. Although the percentages of porosity and average diameter vary slightly among mats, it is possible to observe a relationship of both parameters with the water uptake capacity of the mats. More specifically, the mats with higher porosity have higher water absorption capacity, which would suggest that the wettability properties of these polymeric systems are also affected by changes in the morphology of the mats.”

11- Can the released Ag+ ions be chelated by the functional groups of the amino acids and polymer and re-form Ag NPs on the nanofibers surface?

Indeed, functional groups of amino acids such as amine groups can chelate Ag+ ions. However, PSMA does not contain functional groups that can reduce Ag+ ions to generate AgNPs.

12- Figure 11a&b; what is the reason for inconsistent release of Cal on the 5th and 3rd day, respectively?

It is likely that the distribution of Cal within the fibers is not completely homogeneous, therefore, it is believed that the samples chosen and used on those days belong to a zone of the mesh that contained less amount of Cal

13- Page 21, line 609; how can Ag NPs generate ROS?

AgNPs induce an increase in the generation of intracellular ROS (concentration out of equilibrium) during the process of cellular respiration, with mitochondria being the main site of ROS generation. During oxidative phosphorylation in the inner mitochondrial membrane, oxygen is reduced to water by the addition of electrons in the respiratory chain. Some of these electrons can escape the chain and be accepted by molecular oxygen to form the highly reactive superoxide radical anion (O2−), which is then converted to hydrogen peroxide and, in turn, can be completely reduced to water, or partially to a hydroxyl radical (OH-). The Ag+ ion can increase the rate of superoxide anion production, either by blocking electron transport or by accepting an electron from a respiratory transporter and transferring it to molecular oxygen without inhibiting the respiratory chain, but still increasing radical generation.

Here are some papers that explain the mechanism:

He, D., Garg, S., & Waite, T. D. (2012). H2O2-mediated oxidation of zero-valent silver and resultant interactions among silver nanoparticles, silver ions, and reactive oxygen species. Langmuir, 28(27), 10266-10275.

He, D., Jones, A. M., Garg, S., Pham, A. N., & Waite, T. D. (2011). Silver nanoparticle− reactive oxygen species interactions: application of a charging− discharging model. The Journal of Physical Chemistry C, 115(13), 5461-5468.

Maurer, L. L., & Meyer, J. N. (2016). A systematic review of evidence for silver nanoparticle-induced mitochondrial toxicity. Environmental Science: Nano, 3(2), 311-322.

Round 2

Reviewer 3 Report

The applied revision is fine. The manuscript can be published.